# The Impact of Depression and Anxiety on Adult Cancer Patients’ Health-Related Quality of Life

**DOI:** 10.3390/jcm12062196

**Published:** 2023-03-12

**Authors:** Monira Alwhaibi, Yazed AlRuthia, Ibrahim Sales

**Affiliations:** 1Department of Clinical Pharmacy, College of Pharmacy, King Saud University, Riyadh 11149, Saudi Arabia; yazeed@ksu.edu.sa (Y.A.); isales@ksu.edu.sa (I.S.); 2Pharmacoeconomics Research Unit, College of Pharmacy, King Saud University, Riyadh 11149, Saudi Arabia

**Keywords:** anxiety, cancer, depression, quality of life

## Abstract

Background: Cancer patients are at high risk for mental illness and, in turn, poorer health-related quality of life. This study used nationally representative United States (US) data to examine nuances of the impact of depression and/or anxiety on HRQoL in different cancer groups (e.g., cancer only, cancer and depression, cancer and anxiety, cancer and both conditions). Methods: Adult patients aged 18 years and older with a cancer diagnosis were identified from the Medical Expenditure Panel Survey data for 2012–2016. HRQoL was measured using the SF-12 Physical and Mental Component Summary (PCS & MCS) scores. Multivariate linear regressions were used, controlling for a multitude of factors. Results: Around 12% of the 1712 identified patients with cancer had depression, 13% had anxiety, and 8.4% had both depression and anxiety. Patients with comorbid depression and anxiety had the lowest mean scores of both PCS and MCS compared to patients in other groups. In addition, cancer patients with either depression and/or anxiety were more likely to have lower MCS scores compared to those with cancer only (depression: β = −6.554; anxiety: β = −3.916; both conditions: β = −11.759, *p* < 0.001). Interestingly, patients with comorbid depression and anxiety were more likely to have higher PCS scores compared to those with cancer only. Conclusions: The psychological burden of cancer is immense, with a substantial impact on patients’ HRQoL. Routine screening for depression and anxiety, especially for women and those with low poverty status and comorbidities, should be conducted by healthcare providers to identify those with high odds of having a lower HRQoL. Additionally, early psychiatric interventions, such as psychotherapy and prescription drugs, may positively impact patients’ mental well-being and HRQoL.

## 1. Introduction

Cancer is one of the leading causes of morbidity and mortality worldwide [1]. Early diagnosis and treatment of cancer have resulted in higher survival rates for most types of cancer [2]. In the United States (US), the most common types of newly diagnosed cancer cases in 2019 for women were breast cancer (30%), followed by lung and bronchus cancer (13%). For men, prostate cancer was the most commonly reported type (20%), followed by lung and bronchus cancer (13%) [2]. Cancer diagnosis and treatment are associated with a significant negative psychological impact leading, in many cases, to depression and anxiety. It has been estimated that 8–25% of patients with cancer have depression, and 17% have anxiety [3,4]. However, depression and anxiety are often under-diagnosed and under-treated among cancer patients due to overlapping symptoms with cancer-related fatigue and pain [5].

Undiagnosed and untreated anxiety and depression in patients with cancer can worsen health outcomes and potentially increase cancer-related morbidity and mortality [6,7]. For instance, depression and anxiety among patients with cancer are associated with higher healthcare costs and utilization [8,9,10] and lower adherence to cancer treatment [11,12]. A retrospective analysis of administrative data of 5055 patients with cancer in the US reported that patients with depression have significantly more annual non-mental-health provider healthcare visits, emergency department visits, and hospital readmissions compared to cancer patients without depression [9]. Moreover, depression and anxiety among patients with cancer are also associated with poor health-related quality of life (HRQoL) [13,14]. Among the different domains of HRQoL, the mental and physical domains are believed to be predominately impacted by illnesses such as cancer [15]. Poor HRQoL can negatively affect cancer survival rates. For example, in a cohort of older women with breast cancer, patients’ HRQoL scores were significantly associated with their 10-year risk of mortality [16]. In a cohort of colorectal cancer, patients’ HRQoL, depression, and anxiety scores increased the risk of dying the following year by 22%, 7.2%, and 12%, respectively [17].

Multiple studies have investigated the impact of depression and/or anxiety on HRQoL among patients with breast [18], colorectal [19], lung [20], thyroid [21], and head and neck cancer [22]. These studies suggested that both depression and anxiety are related to worse HRQoL. While these studies contributed to the body of knowledge about the impact of depression and/or anxiety on HRQoL among adults with cancer, none of them compared the impact of depression and/or anxiety on HRQoL using national US data across different groups of cancer patients (e.g., cancer only, cancer and depression, cancer and anxiety, cancer and both conditions). Furthermore, socioeconomic factors, access to care, physical activity, smoking, and medical comorbidities are believed to affect the HRQoL of adults with cancer [15]. Therefore, this study aimed to examine the impact of depression and/or anxiety on HRQoL among different groups of cancer patients (i.e., cancer only, cancer and depression, cancer and anxiety, cancer and both conditions) using nationally representative US data and controlling for a multitude of potential confounders that could impact the HRQoL for adults with cancer. Due to the high rates of comorbid depression and anxiety in patients with cancer, it is crucial to evaluate how these mental health conditions contribute to their health-related quality of life.

## 2. Methods

### 2.1. Study Design and Data

A cross-sectional study was conducted using the Medical Expenditure Panel Survey (MEPS) data. Three alternate years of data (2012, 2014, and 2016) were used based on the recommendations of MEPS to avoid duplicate observations of the same participant (as a result of longitudinal follow-up of MEPS), and multiple years of data were pooled to increase the sample size. The Agency for Healthcare Research and Quality (AHRQ), which is responsible for the MEPS database, makes MEPS data publicly available online for researchers, and data are anonymous. The MEPS is a nationally representative survey of the noninstitutionalized US civilian population with ages ranging from 18 to 64 years old. The MEPS provides information on sociodemographic factors, health insurance, health conditions, medication use, and other health services.

### 2.2. Study Population

Adults aged 18–64 years diagnosed with cancer using the International Classification of Diseases, Ninth Revision, Clinical Modification (ICD-9-CM) clinical diagnoses codes and who were alive during the calendar years were included in the study. The study included adult patients with all types of cancer, such as breast, prostate, colon, rectum, lung, pancreas, lymphomas, etc.

### 2.3. Measures

#### 2.3.1. Outcome: Health-Related Quality of Life (HRQoL)

The MEPS assessed HRQoL using the Short-Form 12 Version 2 (SF-12V2); physical and mental domains of HRQoL were assessed using the Physical Component Summary (PCS) and Mental Component Summary (MCS) of the questionnaire [23]. The PCS and MCS were included in the analysis as continuous variables, with higher scores representing better physical and mental HRQoL, respectively.

#### 2.3.2. Independent Variables

The key independent variable was cancer group; adults with cancer were categorized into four mutually exclusive cancer groups (e.g., cancer only, cancer and anxiety, cancer and depression, or cancer and both conditions). In addition, this study identified individuals with depression and anxiety from the MEPS file using the clinical classification codes (651 for anxiety, 657 for depression) and the ICD-9-CM codes of 296, 300, and 311 for depression. These conditions were first reported by households, recorded by qualified coders, and then converted into clinical classification codes by MEPS researchers. Other independent variables included sociodemographic characteristics (i.e., gender, age in years, race/ethnicity, marital status, region of residence, education level), employment status, poverty status, health insurance (public, private, and uninsured), and medication insurance. Poverty status was defined using family income relative to the federal poverty line (FPL) and classified as poor (<100% FPL), near poor (100% to <200% FPL), middle income (200% to <400% FPL), and high income (≥400% FPL). Physical activity level, smoking status, perceived physical health, and comorbid chronic health conditions were also included as potential confounders. For the purpose of this study, comorbidity refers to one or more other health conditions among persons with an index disease (e.g., cancer) [24].

### 2.4. Statistical Analyses

Descriptive statistics were conducted to describe the study population. Chi-square tests were used to identify the differences in the level of depression or anxiety or both in each independent variable category. Mean differences in HRQoL by cancer group were identified using ANOVA. Multivariable linear regressions were used to examine the association between cancer groups and HRQoL after adjusting in the regression model for age, gender, race/ethnicity, marital status, region of residence, education level, employment status, poverty, health insurance, medication insurance, physical activity, smoking, perceived physical health, and comorbid chronic health conditions. All estimates in the statistical analyses integrated MEPS person-level weights and variance adjustment weights (strata and primary sampling unit) that permit estimates to be nationally representative and account for the MEPS’s complex survey design. SAS 9.4 (SAS Institute Inc., Cary, NC, USA) was used to analyze the data. A *p*-value of less than 0.05 was considered statistically significant.

## 3. Results

### 3.1. Characteristics of the Study Sample

The characteristics of the study sample (1712 adults with cancer) are displayed in Table 1. The majority of the study sample were between 50 and 64 years old (68%), women (60%), employed (65%), and had higher than a high school education (59%). Depression was prevalent in 12% of adults with cancer, 13% had anxiety, and 8.4% had both conditions. Women with cancer had a significantly higher prevalence of depression (13.5% vs. 10%), anxiety (14.8% vs. 10.3%), and comorbid depression and anxiety (10.9% vs. 4.6%) compared to men (*p*-value < 0.001). In addition, unemployed cancer patients had a significantly higher prevalence of depression (18.8% vs. 9.3%) and comorbid depression and anxiety (16.5% vs. 4.1%) compared to the employed (*p*-value < 0.001). Further, adults with cancer and comorbidities (heart, hypertension, asthma, COPD, arthritis, Gastroesophageal Reflux Disease (GERD), etc.) had a significantly higher prevalence of depression, anxiety, and comorbid depression and anxiety compared to those without these comorbidities (*p*-value < 0.001).

### 3.2. Cancer Groups and Health-Related Quality of Life

There was a significant difference in HRQoL (PCS and MCS scores) among cancer groups (Table 2). For example, the lowest mean PCS was observed in cancer patients with both depression and anxiety (mean = 37.9, SE = 1.30) compared to other groups (44.8 for cancer and depression, 48.2 for cancer and anxiety, and 53.2 for cancer only).

### 3.3. Cancer Groups and Health-Related Quality of Life Adjusted Analysis

Table 3 displays the adjusted relationship among cancer groups and HRQoL. A significantly lower HRQoL was found in cancer patients with depression (MCS: β = −6.554, *p* < 0.001), anxiety (MCS: β = −3.916, *p* < 0.001), and both depression and anxiety (MCS: β = −11.759, *p* < 0.001) compared to those with cancer only after adjusting for age, gender, race, marital status, education level, employment status, poverty, insurance, and comorbidities.

## 4. Discussion

This study examined the impact of depression and anxiety on HRQoL among adults with cancer. The study found that about 12% of patients with cancer had depression, 13% had anxiety, and 8.4% had both depression and anxiety. The prevalence rates of both depression and anxiety are similar to the published rates reported in the existing literature and were associated with poor quality of life [3,4].

Our study found a higher rate of depression among women compared to men, unemployed cancer patients compared to employed patients, and patients with comorbidities compared to those without comorbidities. Therefore, special attention should be paid to patients with multiple comorbidities, patients who are unemployed, and female patients to detect early signs of depression and/or anxiety and manage them accordingly.

The HRQoL was measured and compared among four cancer groups (i.e., cancer only, cancer and depression, cancer and anxiety, and cancer and both conditions). The current findings show that neither anxiety nor depression is associated with the physical health domain of HQRoL, and this is consistent with a study of patients with different types and stages of cancer [13]. The study findings also indicate that depression and anxiety are each associated with the mental health domain of HRQoL in patients with cancer. Comparisons of cancer groups suggested that cancer groups with no depression or anxiety had better HRQoL than those with depression only, anxiety only, or comorbid depression and anxiety. Additionally, patients with cancer and comorbid depression and anxiety had worse HRQoL in the mental health domain than cancer patients with either depression or anxiety or without these conditions. This finding indicates the additive effects on HRQoL for adults with cancer with comorbid depression and anxiety. Furthermore, Brown et al., in their study of 405 adult oncology patients, also showed that comorbid anxiety and depression have additive effects on the mental health domains of HRQoL [13].

Our study findings are in line with other internationalg findings from published studies of patients with breast [18], colorectal [19], lung [20], thyroid [21], and head and neck cancer [22]; however, this is the only study that used nationally representative data to evaluate the impact of depression and anxiety (separately and as comorbid conditions) on HRQoL. Previous studies used self-reported scales to measure depression and anxiety; however, this study used clinical diagnostic codes. We also adjusted for comprehensive factors that could affect the association between depression and/or anxiety and HRQoL.

These findings support the need for comprehensive cancer care that includes mental health care and the identification of patients who may benefit from early mental health treatment or social support. Studies have shown that one-third of patients with cancer and depression do not receive any treatment for depression [25,26]. Although anxiety and depression are very common among patients with cancer and have negative health consequences, they are manageable chronic conditions [27]. Therefore, early diagnosis and treatment of depression and anxiety can reduce cancer symptoms and side effects from treatment and improve HRQoL generally. Furthermore, social support has been found to be associated with improvements in anxiety and depression and HRQoL among patients with cancer [28]. This finding supports the need for comprehensive cancer care that includes mental health care and the identification of patients who may benefit from early mental health treatment or social support.

### 4.1. Study Strengths and Limitations

This is the first study that compared the HRQoL among four cancer groups (i.e., cancer only, cancer and depression, cancer and anxiety, and cancer and both conditions). This study used a reliable and valid assessment tool to evaluate the impact of depression and/or anxiety on the HRQoL of patients with cancer. Moreover, this study controlled for comprehensive factors such as sociodemographic characteristics, physical activity, smoking, health insurance coverage, and comorbid chronic health conditions. In addition, using a nationally representative sample of adults with cancer enabled us to obtain accurate estimates of the prevalence rates of depression and/or anxiety among cancer patients. However, the results of this study should be taken in the context of some limitations. Cancer-related factors that may impact the HRQoL, such as stage at cancer diagnosis, cancer treatment, pain, and fatigue symptoms, are unavailable in the MEPS and, therefore, were not adjusted in the analysis. Moreover, the severity of depression and anxiety that may affect the HRQoL of adults with cancer is not available in the MEPS and, hence, was not adjusted for in the regression analysis. Depression and anxiety may be under-recognized, and the prevalence rate of these mental conditions may have been underestimated. Additionally, it was difficult to evaluate the causal relationship due to this study’s cross-sectional design. Finally, the study’s findings cannot be generalized to the elderly (aged 65 years and older) as this study included adults only (18–64 years old).

### 4.2. Clinical Implications

Findings from this study suggest that routine screening for depression and anxiety, especially for women and those with poor poverty status and comorbidities, needs to be performed by oncologists and other healthcare providers to identify those with a depression and anxiety diagnosis and, therefore, high odds of having lower HRQoL andpoorer long-term outcomes. In addition, based on the Pan-Canadian and the American Society of Clinical Oncology (ASCO) guidelines, it is recommended that a periodic assessment of depression and anxiety symptoms be performed for all patients with cancer over the course of care [29]. Depression and anxiety may hinder the management of cancer [30]; therefore, healthcare providers should not delay treating depression and anxiety because early detection and treatment can improve HRQoL and other cancer-related health outcomes [31]. Finally, coordinated care between healthcare providers and the provision of patient-centered information are important elements of patient care and may also improve HRQoL and ameliorate anxiety and depression in patients with cancer [32].

## 5. Conclusions

The results of this study suggest that depression and anxiety are associated with lower HRQoL among patients with cancer. Underestimating the consequences of these psychological distresses may negatively affect cancer health outcomes. Therefore, early assessment and treatment of depression and anxiety by oncologists and other healthcare providers for patients with cancer—especially women, those with poor poverty status, and those with comorbidities—are recommended to alleviate anxiety and depression and, thereby, improve the HRQoL of patients with cancer and other cancer-related health outcomes.

## Figures and Tables

**Table 1 jcm-12-02196-t001:** Characteristics of the study sample (number and weighted percentage). Number and row weighted percentage of characteristics by cancer category among adults with cancer.

	Total Sample		Cancer Only		Cancer and Depression		Cancer and Anxiety		Cancer and Depression and Anxiety		
	N	Wt.%	N	Wt.%	N	Wt.%	N	Wt.%	N	Wt.%	Sig
All	1712	100	1158	66.1	213	12.5	184	13	157	8.4	
Age in years											
18–39	257	12.6	169	64.8	24	10.3	34	15.8	30	9.2	
40–49	340	18.7	231	64.9	37	12.8	41	15.1	31	7.3	
50–64	1115	68.7	758	66.7	152	12.8	109	12	96	8.6	
Gender											
Women	1099	60.6	692	60.8	149	13.5	129	14.8	129	10.9	***
Men	613	39.4	466	74.2	64	10.8	55	10.3	28	4.6	
Race/ethnicity											
White	1040	81	664	64.5	149	13.4	128	13.5	99	8.6	
African American	262	7	197	78.2	30	9.8	18	6.9	17	5.1	
Latino	281	7.3	197	68.7	21	7	28	12	35	12.3	
Other	129	4.7	100	70.8	13	8.7	10	16.3	6	4.3	
Marital status											
Married	992	64.7	725	70.7	108	11.4	100	12.1	59	5.8	***
Wid/Div/Sep	414	20.1	239	56.2	59	13.2	53	15.5	63	15.1	
Never married	306	15.3	194	59.7	46	16.1	31	13.4	35	10.8	
Education level											
LT HS	257	9.1	162	51.4	34	15.8	26	14.1	35	18.7	**
HS	256	14.6	177	61	41	17.9	24	14.4	14	6.6	
>HS	903	58.9	616	68.9	107	11.4	98	12.3	82	7.4	
Missing	296	17.4	203	68.5	31	9.7	36	13.8	26	8.1	
Region											
Northeast	328	19.3	214	68	35	9.9	37	11.9	42	10.3	
Midwest	349	23.1	218	62.5	50	13.2	48	14.9	33	9.5	
South	592	34.2	419	67.3	71	13.5	59	12.2	43	6.9	
West	443	23.4	307	66.2	57	12.3	40	13.3	39	8.2	
Employment											
Employed	1031	65	762	72.3	93	9.3	120	14.3	56	4.1	***
Not employed	681	35	396	54.6	120	18.3	64	10.6	101	16.5	
Poverty status											
Poor	302	11.3	168	52	49	16.5	34	12	51	19.5	***
Near poor	276	12.5	172	52.8	34	16.5	32	15.2	38	15.5	
Middle income	427	23.7	298	65.2	56	15.9	41	10.9	32	8.1	
High income	707	52.5	520	72.7	74	9.1	77	13.7	36	4.5	
Health Insurance											
Private	1170	78.4	847	70.4	122	10.3	124	13.2	77	6.1	***
Public	431	16.5	237	46.9	72	19.4	53	14.5	69	19.3	
Uninsured	111	5.1	74	61.3	19	23.8	7	5.3	11	9.6	
Rx Insurance											
Rx insurance	1060	72.6	778	71.4	105	9.8	113	13	64	5.7	***
No Rx insurance	652	27.4	380	52.1	108	19.4	71	13	93	15.6	
General health											
Ex/very good	713	48.9	560	79.1	62	8.1	65	10.1	26	2.7	***
Good	524	29.3	350	59.1	68	15.7	58	15.9	48	9.2	
Fair/poor	475	21.9	248	46.3	83	17.8	61	15.7	83	20.2	
Physical activity											
≥3 times/week	762	48	566	74.1	73	8.7	79	12.1	44	5.1	***
<3 times/week	948	52	590	58.7	140	15.9	105	13.9	113	11.5	
Smoking											
Current smoker	271	15.1	145	51.2	49	18.6	40	17.8	37	12.4	***
Others	1418	84	1000	69	160	11.3	142	12.2	116	7.5	
Heart											
Yes	271	16.9	137	47.9	46	15.9	41	17.6	47	18.6	***
No	1441	83.1	1021	69.8	167	11.8	143	12.1	110	6.4	
Hypertension											
Yes	657	36.9	394	57.4	104	15.5	84	15.7	75	11.4	***
No	1055	63.1	764	71.2	109	10.7	100	11.5	82	6.7	
Diabetes											
Yes	260	13.8	158	56.9	43	16.1	27	15.5	32	11.5	
No	1452	86.2	1000	67.6	170	11.9	157	12.6	125	7.9	
Hyperlipidemia											
Yes	925	53.4	607	63.5	126	13.4	99	13.2	93	9.9	
No	787	46.6	551	69	87	11.4	85	12.8	64	6.8	
Asthma											
Yes	210	10.6	90	42.8	44	21.2	27	14.3	49	21.7	***
No	1502	89.4	1068	68.9	169	11.4	157	12.9	108	6.9	
COPD											
Yes	309	16.7	142	43.1	62	20.7	39	14.9	66	21.3	***
No	1403	83.3	1016	70.7	151	10.8	145	12.6	91	5.9	
Arthritis											
Yes	716	41.2	388	52.7	122	17.2	91	15.6	115	14.5	***
No	996	58.8	770	75.4	91	9.1	93	11.2	42	4.2	
GERD											
Yes	260	15.8	128	47.8	46	21.1	37	14.9	49	16.2	***
No	1452	84.2	1030	69.5	167	10.8	147	12.7	108	7	

Note: Based on 1712 adults with cancer, aged > 18 years. Asterisks (*) represent significant differences in cancer groups from chi-square tests. Wt: weighted; LT: less than; GERD: Gastroesophageal Reflux Disease; GT: greater than; Rx: medication; Wid/Div/Sep: widowed, divorced, and separated. *** *p* < 0.001; ** 0.001 ≤ *p* < 0.01.

**Table 2 jcm-12-02196-t002:** Weighted means and standard errors of HRQoL scores by cancer groups. Adults with cancer, Medical Expenditure Panel Survey 2012–2016.

	Total Sample		Cancer Only		Cancer and Depression		Cancer and Anxiety		Cancer and Depression and Anxiety		
	Mean	SD	Mean	SE	Mean	SE	Mean	SE	Mean	SE	Sig.
HRQoL											
PCS	49.03	10.96	53.21	0.26	44.86	0.95	48.20	1.13	37.95	1.30	***
MCS	44.97	12.43	48.30	0.46	42.06	1.28	44.06	1.63	38.84	1.26	***

Note: Based on 1712 adults with cancer, aged > 18 years. Significant mean differences in HRQoL for cancer groups. HRQoL: health-related quality of life; MCS: Mental Component Summary; PCS: Physical Component Summary; SE: standard error; SD: standard deviation; Sig: significance. *** *p* < 0.001.

**Table 3 jcm-12-02196-t003:** Parameter estimates and standard errors from multivariate linear regressions on HRQoL among adults with cancer, Medical Expenditure Panel Survey 2012–2016.

	Health-Related Quality of Life					
	PCS			MCS		
	β	SE	Sig.	β	SE	Sig.
Cancer group						
Cancer and depression	0.604	0.518		−6.554	0.583	***
Cancer and anxiety	−0.718	0.432		−3.916	0.845	***
Cancer and depression and anxiety	1.784	0.399	***	−11.759	0.634	***
Cancer only (Ref.)						
Age in years						
18–39	1.892	0.4	***	−1.599	0.493	**
40–49	1.157	0.307	***	−1.413	0.472	**
50–64 (Ref.)						
Gender						
Women	0.534	0.212	*	0.093	0.332	
Men (Ref.)						
Race/ethnicity						
White	1.497	0.49	**	0.291	0.453	
African American	1.197	0.478	*	0.12	0.57	
Latino	1.439	0.567	*	0.661	0.694	
Other (Ref.)						
Marital status						
Married	−1.512	0.454	**	1.882	0.541	***
Wid/Div/Sep	−0.631	0.572		0.627	0.656	
Never married (Ref.)						
Education level						
LT HS	−0.808	0.307	**	−0.183	0.373	
HS	−1.438	0.358	***	1.094	0.59	
>HS	0.964	0.509		0.125	0.68	
Missing (Ref.)						
Region						
Northeast	0.295	0.362		0.054	0.51	
Midwest	−1.391	0.336	***	0.122	0.544	
South	−1.233	0.304	***	−1.294	0.456	**
West (Ref.)						
Employment						
Employed	3.642	0.436	***	−0.234	0.273	
Not employed (Ref.)						
Poverty status						
Poor	−0.497	0.639		−1.227	0.592	*
Near poor	−0.498	0.617		−1.876	0.574	**
Middle income	−1.306	0.299	***	−0.74	0.33	*
High income (Ref.)						
Health Insurance						
Private	2.922	0.393	***	0.367	0.607	
Public	−0.568	0.637		−0.767	0.667	
Uninsured (Ref.)	0	0		0	0	
Rx Insurance						
Rx insurance	0.123	0.453		1.809	0.423	***
No Rx insurance (Ref.)						
General health						
Ex/very good	12.876	0.389	***	4.804	0.41	***
Good	7.707	0.374	***	3.416	0.492	***
Fair/poor (Ref.)						
Physical activity						
≥3 times/week	0.645	0.227	**	1.363	0.255	***
<3 times/week (Ref.)						
Smoking						
Current smoker	−1.133	0.343	**	−0.721	0.62	
Heart						
Yes	−0.646	0.367		−0.798	0.46	
Hypertension						
Yes	−1.495	0.381	***	1.634	0.313	***
Diabetes						
Yes	−2.384	0.449	***	−0.344	0.583	
Hyperlipidemia						
Yes	0.803	0.23	***	−0.721	0.302	*
Asthma						
Yes	−0.608	0.718		−0.615	0.723	
COPD						
Yes	−2.156	0.489	***	0.711	0.698	
Arthritis						
Yes	−4.613	0.343	***	−0.356	0.373	
GERD						
Yes	−3.778	0.384	***	0.757	0.395	

Note: Based on 1712 adults with cancer, aged 18 years and older. Asterisks denote statistical significance in parameter estimates from multivariate linear regressions on health-related quality of life. *** *p* < 0.001; ** 0.001 < *p* < 0.01; * 0.01 < *p* < 0.05. LT: less than; GERD: Gastroesophageal Reflux Disease; GT: greater than; Rx: medication; Ref: reference group; SE: standard error; Sig: significance. Wid/Div/Sep: widowed, divorced, and separated.

## Data Availability

The dataset used in this article is available from MEPS database and is openly made available for researchers at the following website: https://meps.ahrq.gov/data_stats/download_data_files.jsp (accessed on 26 December 2021). The Agency for Healthcare Research and Quality makes MEPS data publicly available online for researchers, and data are anonymous.

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
