# Peer review of "The Impact of Depression and Anxiety on Adult Cancer Patients’ Health-Related Quality of Life"

_jcm, 2023, doi:10.3390/jcm12062196_

Round 1
Reviewer 1 Report
Thank you for the opportunity to review this manuscript. The authors describe an important study examining the nuanced impact of depression and/or anxiety on health-related quality of life among adult cancer patients. The authors also explore the role of socioeconomic characteristics as confounding factors. The primary finding was that patients with multiple comorbidities, who are female or unemployed experience higher rates of depression, anxiety, and/or comorbid depression and anxiety. Comparisons between cancer groups (i.e., cancer only, cancer and depression, cancer and anxiety, or cancer and comorbid depression and anxiety) found that patients with cancer and comorbid depression and anxiety experienced the worst health-related quality of life in the mental health domains. Implications point to the need for routine depression and anxiety screening for cancer patients, given the potential impact of poorer health-related quality of life for long-term health outcomes (e.g., morbidity).
I have some minor suggestions for improving the manuscript. I think that the abstract could do a better job of describing the study. For example, I think it’s important to highlight that you’ve used nationally representative data within the abstract, given that this is a strength of your study. You might consider changing the Background section of the Abstract to: “Cancer patients are at high risk of mental illness, and in turn poorer health-related quality of life. This study used nationally representative US data to examine nuances in the impact of depression and/or anxiety on HRQoL in different cancer groups (e.g., cancer only, cancer and depression, cancer and anxiety, cancer and both conditions).”
I think it would also be useful to define the cancer groups in your Abstract given that you report your Results as comparisons between groups – the groups currently aren’t defined in your Abstract (e.g., cancer only, cancer and depression, cancer and anxiety, cancer and both conditions).
Introduction:
- Consider rewording sentence in lines 38-40: “However, depression and anxiety are often under-diagnosed and under-treated among cancer patients due to overlapping symptoms with cancer-related fatigue and pain.”
- Consider rewording sentence in lines 43-44: “For instance, depression and anxiety among patients with cancer are associated with lower adherence to cancer treatment and higher service utilisation.”
Methods:
- Query: Is the MEPS data publicly available? If so, consider providing a reference. Consider adding an explanation that this is publicly available data – then it is clear that there is no need for ethical approval and that patients were not directly involved in the study.
- Suggestion: Consider adding some information about the people who completed the MEPS to Section 2.2 – for example, age range?
- Query: Is the statement in Section 2.3 necessary? Consider removing this if not a requirement of the journal.
- Suggestion: Consider rewording first sentences in Section 2.4.1: “The MEPS assessed HRQoL using the Short-Form 12 Version 2. Physical and mental domains of HRQoL were assessed using the Physcial Component Summary and Mental Component Summary of the questionnaire.”
- Query: I’m a bit unclear on how individuals with depression and anxiety were identified – was this self-reported or assessed by a clinician filling out the survey on behalf of respondent? Perhaps would be useful to include the actual question used in the survey – for example, did the survey ask “have you ever been diagnosed with a mental health condition?”
- Suggestion: Consider simplifying the information included in Section 2.4.2 – I don’t think it’s necessary to include all of the categories. Consider rewording lines 102-115: “Other independent variables included sociodemographic characteristics, including gender, age, race/ethnicity, marital status, region of residence, education level, health insurance, medication insurance, employment status, and poverty status. Poverty status was defined using... Physical activity level, smoking status, alcohol use, perceived physical health, and comorbid chronic health conditions were also included as potential confounders. For the purpose of this study, comorbidity refers to one or more other health conditions among persons with an index-disease (e.g., cancer).”
- Suggestion: Consider changing “risk of” in line 118 to “level of”.
- Check that the variables you've included in your independent variables section match what you've included in statistical analyses section (lines 121-123).
Results:
- Suggestion: Change “Besides” to “further” (line 139).
Discussion:
- Make sure you're consistent in how you refer to the comorbid depression and anxiety group (in at least one place (line 181), you use "depression-anxiety").
- Suggestion: Consider rewording sentence in lines 178-181: “Comparisons of cancer groups suggested that cancer groups with no depression or anxiety had between HRQoL than those with depression only, anxiety only, or comorbid depression and anxiety.”
- Suggestion: Consider changing “the worst” in line 182 to “worse” or “poorer”.
- Suggestion: Consider connecting the two sentence in lines 188-191: "Although our study findings are in line with some published studies among patients with breast, colorectal, lung, thyroid, and head and neck cancers, this is the only study that has used nationally representative data to evaluate the impact of depression and anxiety (separately and as comorbid conditions) on HRQoL."
- Suggestions: Consider rewording sentence in lines 195-197: "This finding supports the need for comprehensive cancer care that includes mental health care and the identification of patients who may benefit from early mental health treatment or social support."
- Suggestion: Consider rewording sentence in lines 200-201: “Early diagnosis and treatment and depression and anxiety can reduce cancer symptoms and side effects from treatment and improve HRQoL generally.”
- In lines 202-203, you state that social support has been found to be associated with improvement in anxiety and depression – what about HRQoL?
- Suggestion: Consider rewording sentence in lines 224-226: “Findings from this study propose that routine screening for depression and anxiety, especially for women and those with poverty status and comorbidities, is needed by oncologists and other healthcare providers to identify those who may be at risk of lower HRQoL and therefore poorer longterm outcomes.”
- Query: When you say “low poverty status”, do you mean that these participants experience higher levels of participants? Consider rewording on line 225. Could reword as “those with ‘poor’ poverty status”.
- I think it would also be useful to include the above finding (routine screening for women, those with poverty status and comorbidities) in abstract.
Author Response
We want to thank the editor and the reviewers for the time and effort they spent reviewing our manuscript entitled “The Impact of Depression and Anxiety on Adult Cancer Patients’ Health-Related Quality of Life", their valuable and insightful comments have improved our manuscript substantially.
We are excited to have been allowed to revise our manuscript and respond to the minor revisions. We have gone through all comments received, and appropriate changes/amendments have been made correspondingly to the paper (Highlighted) are summarized in the following:
Reviewer(s)' Comments to the Authors
Reviewer# 1
Abstract
I have some minor suggestions for improving the manuscript. I think that the abstract could do a better job of describing the study. For example, I think it’s important to highlight that you’ve used nationally representative data within the abstract, given that this is strength of your study. You might consider changing the Background section of the Abstract to: “Cancer patients are at high risk of mental illness, and in turn poorer health-related quality of life. This study used nationally representative US data to examine nuances in the impact of depression and/or anxiety on HRQoL in different cancer groups (e.g., cancer only, cancer and depression, cancer and anxiety, cancer and both conditions).”
I think it would also be useful to define the cancer groups in your Abstract given that you report your Results as comparisons between groups – the groups currently aren’t defined in your Abstract (e.g., cancer only, cancer and depression, cancer and anxiety, cancer and both conditions).
Response: Thank you so much for this suggestion. We have modified the abstract according to this suggestion (lines 14-17). In addition, we thank the reviewer for his valuable comments that have improved our manuscript substantially.
Introduction
Comment # 1: Consider rewording sentence in lines 38-40: “However, depression and anxiety are often under-diagnosed and under-treated among cancer patients due to overlapping symptoms with cancer-related fatigue and pain.”
Response: Thank you so much for this suggestion. We have made the suggested rewording (lines 45-47).
Comment # 2: Consider rewording sentence in lines 43-44: “For instance, depression and anxiety among patients with cancer are associated with lower adherence to cancer treatment and higher service utilisation.”
Response: Thank you for this suggestion; we have reworded this sentence (lines 51-53).
Methods
Comment # 3: Query: Is the MEPS data publicly available? If so, consider providing a reference. Consider adding an explanation that this is publicly available data – then it is clear that there is no need for ethical approval and that patients were not directly involved in the study.
Response: The Agency for Healthcare Research and Quality, which is responsible for the MEPS database, makes MEPS data publicly available data online for researchers and data are not restricted since it is an anonymous data. Link for AHRQ website: https://www.ahrq.gov/cpi/about/otherwebsites/meps.ahrq.gov/index.html
We have added an explanation to the manuscript (lines 89-91).
Comment # 4: Suggestion: Consider adding some information about the people who completed the MEPS to Section 2.2 – for example, age range?
Response: Thanks for this suggestion; we have added this information to section 2.1 (lines 91-93).
Comment # 5: Query: Is the statement in Section 2.3 necessary? Consider removing this if not a requirement of the journal.
Response: It is not a requirement of the JCM journal; therefore, we have removed it and renumbered the sections
Comment # 6: Suggestion: Consider rewording first sentences in Section 2.4.1: “The MEPS assessed HRQoL using the Short-Form 12 Version 2. Physical and mental domains of HRQoL were assessed using the Physcial Component Summary and Mental Component Summary of the questionnaire.”
Response: Thank you for this suggestion; we have reworded this section (lines 106-111).
Comment # 7: Query: I’m a bit unclear on how individuals with depression and anxiety were identified – was this self-reported or assessed by a clinician filling out the survey on behalf of respondent? Perhaps would be useful to include the actual question used in the survey – for example, did the survey ask “have you ever been diagnosed with a mental health condition?”
Response: We have described it in section 2.4.2 “These conditions were first reported by households, recorded by qualified coders, and then converted into clinical classification codes (ICD-9-CM codes) by MEPS researchers (lines 121-123).
Comment # 8: Suggestion: Consider simplifying the information included in Section 2.4.2 – I don’t think it’s necessary to include all of the categories. Consider rewording lines 102-115: “Other independent variables included sociodemographic characteristics, including gender, age, race/ethnicity, marital status, region of residence, education level, health insurance, medication insurance, employment status, and poverty status. Poverty status was defined using... Physical activity level, smoking status, alcohol use, perceived physical health, and comorbid chronic health conditions were also included as potential confounders. For the purpose of this study, comorbidity refers to one or more other health conditions among persons with an index-disease (e.g., cancer).”
Response: Thank you for this suggestion; we have simplified this section (lines 123-138).
Comment # 9: Suggestion: Consider changing “risk of” in line 118 to “level of”.
Response: Suggested change has been made (line 142).
Comment # 10: Check that the variables you've included in your independent variables section match what you've included in statistical analyses section.
Response: Thank you for this correction; we have corrected it in the statistical analyses section (lines 146-148).
Results:
Comment # 11: Suggestion: Change “Besides” to “further” (line 139).
Response: Suggested change has been made (lines 164).
Discussion:
Comment # 12: Make sure you're consistent in how you refer to the comorbid depression and anxiety group (in at least one place (line 181), you use "depression-anxiety").
Response: we have revised this to ensure the consistency of reporting.
Comment # 13: Suggestion: Consider rewording sentence in lines 178-181: “Comparisons of cancer groups suggested that cancer groups with no depression or anxiety had between HRQoL than those with depression only, anxiety only, or comorbid depression and anxiety.”
Response: Thank you so much for this suggestion. We have made the suggested rewording (lines 205-207).
Comment # 14: Suggestion: Consider changing “the worst” in line 182 to “worse” or “poorer”.
Response: Suggested change has been made (line 208).
Comment # 15: Suggestion: Consider connecting the two sentence in lines 188-191: "Although our study findings are in line with some published studies among patients with breast, colorectal, lung, thyroid, and head and neck cancers, this is the only study that has used nationally representative data to evaluate the impact of depression and anxiety (separately and as comorbid conditions) on HRQoL."
Response: thank you for this comment. We have now connected these two sentences (lines 216-220).
Comment # 16: Suggestions: Consider rewording sentence in lines 195-197: "This finding supports the need for comprehensive cancer care that includes mental health care and the identification of patients who may benefit from early mental health treatment or social support."
Response: Thank you for this suggestion; we have reworded this sentence per the reviewer's comment (lines 224-226).
Comment # 17: Suggestion: Consider rewording sentence in lines 200-201: “Early diagnosis and treatment and depression and anxiety can reduce cancer symptoms and side effects from treatment and improve HRQoL generally.”
Response: Thank you for this suggestion; we have reworded this sentence (lines 231-233).
Comment # 18: In lines 202-203, you state that social support has been found to be associated with improvement in anxiety and depression – what about HRQoL?
Response: Thank you for this suggestion; we have cited two studies that support the relationship between social support and HRQoL (lines 235-237).
Comment # 19: Suggestion: Consider rewording sentence in lines 224-226: “Findings from this study propose that routine screening for depression and anxiety, especially for women and those with poverty status and comorbidities, is needed by oncologists and other healthcare providers to identify those who may be at risk of lower HRQoL and therefore poorer longterm outcomes.”
Response: Thank you for this suggestion; we have reworded this sentence (lines 259-262).
Comment # 20: Query: When you say “low poverty status”, do you mean that these participants experience higher levels of participants? Consider rewording on line 225. Could reword as “those with ‘poor’ poverty status”.
Response: We have reworded ‘low’ to ‘poor’ per the reviewer's comment (line 260).
Comment # 21: I think it would also be useful to include the above finding (routine screening for women, those with poverty status and comorbidities) in abstract.
Response: Thank you for this suggestion; we have added this to the abstract (lines 29-33).

Reviewer 2 Report
While ‘The Impact of Depression and Anxiety on Adult Cancer Patients’ Health-Related Quality of Life’ is definitely an interesting work, its authors might want to consider elaborating on the following:
1) It would be wise to explain why the authors used MEPS data for 2012, 2014 or 2016 (the data obtained almost 11 years ago) and they didn’t use the data for 2013 and 2015.
2) Were the authors granted permission to use the SF-12 questionnaire, which, otherwise, is protected by copyright?
3) The authors should explain why the study population was divided into such age groups.
4) The Discussion section should contain information on how the study results compare to global findings.
Author Response
We want to thank the editor and the reviewers for the time and effort they spent reviewing our manuscript entitled “The Impact of Depression and Anxiety on Adult Cancer Patients’ Health-Related Quality of Life", their valuable and insightful comments have improved our manuscript substantially.
We are excited to have been allowed to revise our manuscript and respond to the minor revisions. We have gone through all comments received, and appropriate changes/amendments have been made correspondingly to the paper (Highlighted) are summarized in the following:
Reviewer(s)' Comments to the Authors
Reviewer# 2
GENERAL COMMENTS
While ‘The Impact of Depression and Anxiety on Adult Cancer Patients’ Health-Related Quality of Life’ is definitely an interesting work, its authors might want to consider elaborating on the following:
Comment # 1: It would be wise to explain why the authors used MEPS data for 2012, 2014 or 2016 (the data obtained almost 11 years ago) and they didn’t use the data for 2013 and 2015.
Response: Thanks for your valuable comment; three alternate years of data (2012, 2014, and 2016) were used based on the recommendations of MEPS to avoid duplicate observations of the same participant (as a result of longitudinal follow-up of MEPS) 1 and multiple years of data were pooled to increase the sample size. Now we have explained it in the methods (lines 86-89).
Comment # 2: Were the authors granted permission to use the SF-12 questionnaire, which, otherwise, is protected by copyright?
Response: Thank you for pointing this out, we have used MEPS data which is publicly available and have not used a survey or collected data in this study.
Comment # 3: The authors should explain why the study population was divided into such age groups.
Response: Thank you for this suggestion; we have added the explanation to the introduction part (lines 79-62).
Comment # 4: The Discussion section should contain information on how the study results compare to global findings.
Response: Thank you for pointing this out; we have now compared our results to the global findings (lines 216-223).
- Sommers JP. An examination of state estimates using multiple years of data from the medical expenditure panel survey, household component: Agency for Healthcare Research and Quality; 2006.
